# Factors mediating the pressor response to isometric muscle contraction: An experimental study in healthy volunteers during lower body negative pressure

**Niels A. Stens**[1,2], **Jonny Hisdal**[2,3], **Espen F. Bakke**[4], **Narinder Kaur**[2,5], **Archana Sharma**[6,7], **Einar Stranden**[2], **Dick H. J. Thijssen**[1,8], **Lars Øivind Høiseth**[7] *

1 Department of Physiology, Research Institute for Health Sciences, Radboud University Medical Center, Nijmegen, The Netherlands, 2 Department of Vascular Surgery, Division of Cardiovascular and Pulmonary Diseases, Section of Vascular Investigations, Oslo University Hospital, Oslo, Norway, 3 Institute of Clinical Medicine, Faculty of Medicine, University of Oslo, Oslo, Norway, 4 Institute of Aviation Medicine, Norwegian Armed Forces Medical Service, Oslo, Norway, 5 Dermatology Center Telemark, Porsgrunn, Norway, 6 Department of Endocrinology, Oslo University Hospital, Oslo, Norway, 7 Department of Anesthesiology, Oslo University Hospital, Oslo, Norway, 8 Research Institute for Sport and Exercise Sciences, Liverpool John Moores University, Liverpool, United Kingdom

* Lars.Oivind.Hoiseth@hotmail.com

**Data Availability Statement:** All relevant data are within the manuscript and its Supporting Information files.

## Abstract

Whilst both cardiac output (CO) and total peripheral resistance (TPR) determine mean arterial blood pressure (MAP), their relative importance in the pressor response to isometric exercise remains unclear. This study aimed to elucidate the relative importance of these two different factors by examining pressor responses during cardiopulmonary unloading leading to step-wise reductions in CO. Hemodynamics were investigated in 11 healthy individuals before, during and after two-minute isometric exercise during lower body negative pressure (LBNP; -20mmHg and -40mmHg). The blood pressure response to isometric exercise was similar during normal and reduced preload, despite a step-wise reduction in CO during LBNP (-20mmHg and -40mmHg). During -20mmHg LBNP, the decreased stroke volume, and consequently CO, was counteracted by an increased TPR, while heart rate (HR) was unaffected. HR was increased during -40 mmHg LBNP, although insufficient to maintain CO; the drop in CO was perfectly compensated by an increased TPR to maintain MAP. Likewise, transient application of LBNP (-20mmHg and -40mmHg) resulted in a short transient drop in MAP, caused by a decrease in CO, which was compensated by an increase in TPR. This study suggests that, in case of reductions of CO, changes in TPR are primarily responsible for maintaining the pressor response during isometric exercise. This highlights the relative importance of TPR compared to CO in mediating the pressor response during isometric exercise.

**Funding:** The author received no specific funding for this work.

**Competing interests:** The authors have declared that no competing interests exist.

## Introduction

Isometric handgrip exercise is known to elicit increases in mean arterial blood pressure (MAP) [1–3]. Although MAP is well-regulated during resting conditions, the increased intra-muscular pressure during isometric exercise causes an increased circulation to the working muscle. The cardiovascular response to isometric exercise is the outcome of an interaction between several factors, including central command, afferent input from skeletal muscle receptors, and arterial and cardiopulmonary baroreceptors [4]. Despite the powerful response of this mechanism being observed a decade ago, its hemodynamic mechanisms remain to be fully elucidated. It remains debatable whether the pressor response is due to increases in cardiac output (CO) [5–9], total peripheral resistance (TPR) [10–14] or both [15–18]. CO changes during IHG is primarily driven by an elevated heart rate (HR), whilst the stroke volume (SV) is slightly reduced due to the tachycardia and increased afterload, or even maintained following both augmented ventricular contractility [19, 20] and constant or elevated preload via central blood volume mobilization [21].

Interestingly, the blood pressure response to a single session of isometric exercise relates to the future risk of developing hypertension [22, 23] and the magnitude of resting blood pressure lowering following chronic exposure [24, 25]. Knowledge of acute blood pressure regulation is therefore clinically important in e.g. hypertensive individuals with abdominal aneurysms who are advised to avoid activities causing a high blood pressure [26]. Nevertheless, studies indicate that isometric exercise may be used for the prevention and treatment of cardiovascular disease [27, 28].

Several studies have shown that the exercise pressor response [29–31], the sympathetic nerve response [29, 32, 33] and the increase in MAP [11] in response to isometric exercise are essentially unchanged following cardiopulmonary unloading as induced by lower body negative pressure (LBNP). It is generally agreed that mild LBNP (-20mmHg) reduces central venous pressure (CVP) and SV while HR remains unaffected. Despite a fall in CO due to a reduced SV, MAP is normally well-maintained through an increased TPR [11, 34–39]. During moderate LBNP -40mmHg, CVP is decreased even further and HR normally increased by 20–30% compared to rest [39, 40].

The present study was designed to elucidate in detail the circulatory mechanisms by which the pressor response is maintained during cardiopulmonary unloading with different levels of reduced SV induced by LBNP, both prior to and during isometric muscle contraction. To illuminate the role of SV, HR and TPR in the blood pressure response to isometric exercise, we have followed beat-by-beat all these hemodynamic parameters involved in the regulation of MAP during isometric handgrip, during continuous and transient application of –20 and –40 mmHg LBNP, during an ongoing contraction as well as reduction in preload with –20 and –40 mmHg LBNP prior to isometric exercise (**Fig 1**).

We hypothesized that in situations with reduced CO due to reduced preload and SV, the pressor response is maintained through an increase in TPR.

## Materials and methods

### Subjects

Eleven healthy volunteers (five males) were recruited for this study (mean (SD), age 23.6 (5.6) years, height 171.1 (9.0) cm, weight 67.8 (10.3) kg). All subjects were non-medicated, non-smokers, normotensive (BP <140/90 mmHg), and had no history of cardiovascular or pulmonary disease. All participants were asked to refrain from drinking coffee/tea on the experimental day, and exercising or eating in the two hours before the start of the experiment.

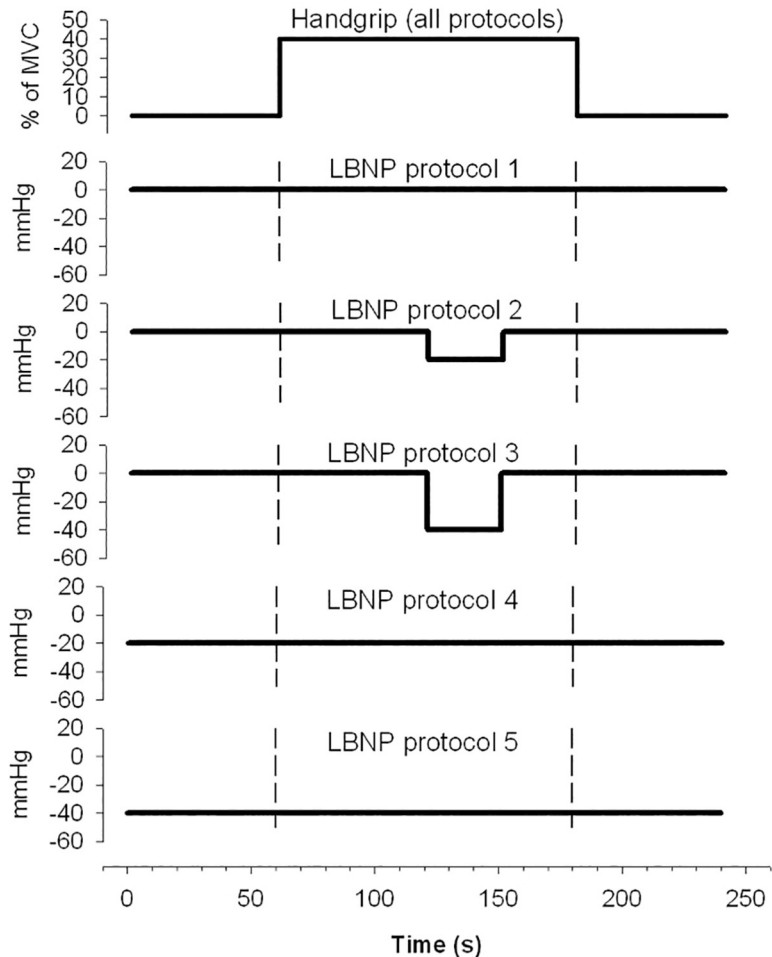

**Fig 1. Handgrip and application of LBNP during the different protocols.**

The procedures were in accordance with institutional guidelines and conformed to the declaration of Helsinki. The study was approved by the regional ethics committee (Reference-116-03042; REK sør-øst, Pb 1130, Blindern, 0318 Oslo, Norway; https://rekportalen.no/). Participants gave written and verbal informed consent before participation.

## Experimental design

In this experimental study, each subject participated in five different protocols (P1-P5; **Fig 1**). The start of the first experiment was preceded by individual familiarization with the experiment, weight and height determination, and 30 min acclimation in supine position. The experiments were carried out in the order P1 to P5, and to avoid any influence of the major hemodynamic effects of LBNP on the results of subsequent protocols, each protocol was followed by 5–7 minutes of intermission. Each protocol was repeated four times, and consisted of four minutes, divided into one-minute baseline, two-minute isometric handgrip and one-minute recovery. Isometric handgrip was accompanied by either transient (P2, P3) or continuous (P4, P5) mild LBNP (-20mmHg) or moderate LBNP (-40mmHg), with P1 being the control with a normal preload. In P4 and P5, LBNP was applied 1 min before the recording started. Subjects were lightly clothed and lay comfortably on a bench in supine position during the test

runs. The ambient temperature in the room was between 23–24˚C and the subjects were in their thermoneutral zone [41]. All experiments were performed at the same time of day to control for diurnal variation.

## Lower body negative pressure

LBNP (-20 mmHg and -40 mmHg) was applied by a custom-built chamber and pressure control system (E. Stranden) designed to introduce precise and rapid changes in LBNP, as previously described elsewhere [42]. The lower body was inside the LBNP chamber and sealed at the level of the iliac crest. During rapid onset and release, LBNP was both reduced and returned to ambient pressure in less than 0.3 s. The onset and release of LBNP were induced so that the pressure profile in the LBNP chamber was identical for all tests. **Fig 1** shows the chamber pressure during the different protocols.

## Handgrip

A custom made handgrip aperture was used to record and display the force exerted by the test subjects as they gripped the handle with their right hand. A visual display gave the subjects continuous information, enabling them to maintain the intended force. Throughout all protocols, the test subjects were asked to exert a force corresponding to 40% of their individual and previously calculated maximal voluntary contraction force (MVC). MVC was determined approximately 10 min prior to the experimental session by asking the test subject to press with maximal force around the handgrip aperture for a three-second period. The mean force exerted in three such sessions was calculated and used as MVC. During the two-minute isometric exercise periods, the participants were instructed to avoid the Valsalva manoeuvre and to relax all the muscles not primarily involved in contraction to avoid recruitment of accessory muscle mass.

## Measurements

Beat-to-beat stroke volume (SV) was recorded by the ultrasound Doppler method (SD-50; GE Vingmed Ultrasound, Horten, Norway) using a 2 MHz suprasternal probe [43]. Instantaneous HR was obtained from the duration of each R-R interval of the ECG signal (SD-50). Beat-to-beat CO was calculated from the corresponding HR and SV values. Blood flow velocity in the brachial artery of the left (resting) arm (BBFV) was measured using ultrasound Doppler with an operating frequency of 10 MHz. The circular transducer was held by the operator over the cubital fossa with the ultrasound beam directed towards the brachial artery. The instantaneous cross-sectional mean velocity was calculated by the SD-50 and transferred online to the computer for beat-by-beat time averaging, gated by the R waves of the electrocardiogram.

Laser Doppler (Periflux PF 4000; Perimed AB, Järnfälla, Sweden) was used to measure acral skin blood flow perfusion (ASBF) in the pulp of the left index finger at a sampling frequency of 2 Hz. The laser Doppler probes were fastened to the skin with narrow double-sided tape (Kontron Instruments, Ltd, UK). Finger arterial pressure of the left middle finger was continuously recorded by a photoplethysmographic device (2300 Finapres BP monitor; Ohmeda, Madison, Wis., USA). Care was taken to adjust the arm so that the finger was at heart level. The instantaneous pressure output was transferred online to the recording computer where beat-to-beat MAP was calculated by numerical integration. This method has shown to provide MAP values in good accordance with intra-arterial pressure [44, 45].

Local peripheral resistance (LPR) was calculated as (MAP/BBFV). TPR was calculated as (MAP/CO), where MAP was used as an approximation to the perfusion pressure across the systemic circulation and CO as an estimate for averaged flow through the resistance vessels.

For ethical considerations, we did not measure central venous pressure. However, the effect of different levels of LBNP on CVP is previously described [40].

## Data analysis

Blood flow velocity in the ascending aorta and the brachial artery were sampled at 50 Hz. HR was sampled beat-by-beat, while SV, CO, MAP, TPR and LPR were calculated for each heartbeat. To allow for analysis, all recorded variables were converted into a 2-Hz sampled signal by interpolation. Throughout the recording period, considerable beat-to-beat variation was present in the recorded variables. This variation has been reported by other authors [46, 47] and is partly due to the influence of respiration [46, 48]. For each subject, variations in the recorded variables not related to the pressor response or to the onset and release of LBNP were partly eliminated by the coherent averaging technique, comprising the calculation of the average response from four identical runs for every test protocol. Finally, the individual average curves from all 11 subjects were pooled and used to calculate the inter-individual averaged responses for the five protocols by finding the mean value in each set of synchronous samples for each 2-Hz time step, presented in Figs 2 and 3.

## Statistical analysis

Values are reported as mean (standard deviation), unless otherwise stated. To explore the effect of IHG during continuous LBNP, the median of the first minute (0 s to 60 s; baseline, without IHG) was compared to that of the last 30 s of IHG (150 s to 180 s) in P1, P4 and P5. To explore the effect of transient LBNP, the first minute was compared to the last 20 s of transient LBNP (130 s to 150 s) during IHG in P1, P2 and P3. Calculations were performed by entering the median values as response variables in linear mixed regression models with subject as a random effect. LBNP-level (0, 20 and 40 mmHg) and IHG/ no-IHG were entered as explanatory factors with interaction effects to describe effects of different LBNP on the response to IHG and *vice versa*. Calculations were performed in R 3.6.3 [49] and RStudio 1.2.5042 [50] using the *nlme* [51] and *multcomp* [52]-packages. Regression model assumptions were checked by visually inspecting Q-Q-plots and standardized residuals vs. predicted values. P-values < 0.05 were considered statistically significant.

## Results

Detailed development of the pooled average responses of the different cardiovascular variables in protocol P1, P4 and P5 is depicted in **Fig 2**. No significant differences were observed in the blood pressure response (interaction effects in Table 1) during reduced preload induced by mild (-20mmHg) or moderate LBNP (-40mmHg) prior to the exercise period despite a significant reduction in CO (Table 1). In addition, a decrease in SV during IHG was offset by an increase in HR, thereby maintaining CO in the absence of LBNP. Based on the interaction effects, LBNP did not significantly affect the response to IHG in MAP or CO (Table 1). The observed gradual increase in blood pressure can primarily be explained by the gradual increase in TPR. During application of mild (-20mmHg) and moderate LBNP (-40mmHg), resting SV dropped with 17% (from 59 to 49 ml/beat) and 32% (from 59 to 40 ml/beat) respectively. With the addition of isometric handgrip, SV showed a similar decrease in both mild (-20mmHg) and moderate LBNP (-40mmHg) compared with isometric handgrip during normal preload (no significant interaction effects). As expected, mild LBNP (-20 mmHg) did not affect HR either before, during or after isometric handgrip, while moderate LBNP (-40 mmHg) increased HR with about 7.2% in rest. The HR response to isometric handgrip during mild LBNP (-20 mmHg) showed a similar trend as during isometric handgrip without application

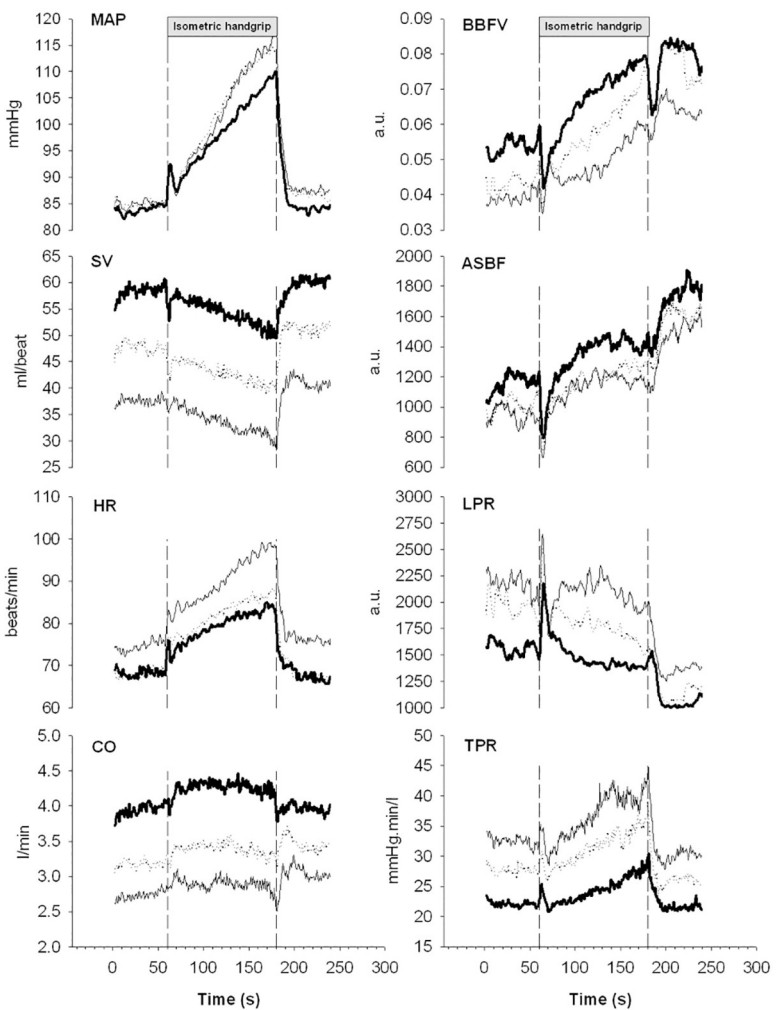

**Fig 2. Pooled average trends of the individual cardiovascular responses (n = 11) to 2 minutes of isometric handgrip contraction (40% of maximal voluntary contraction force).** Bold lines show mean arterial pressure (MAP), stroke volume (SV), heart rate (HR), cardiac output (CO), blood flow velocity in the brachial artery (BBFV), acral skin blood flow perfusion (ASBF), local peripheral resistance (LPR) and total peripheral resistance (TPR) during isometric handgrip in supine position (P1). Dotted lines show the responses during reduced preload induced by mild lower body negative pressure (LBNP -20 mmHg, P4)), and hairlines the responses during LBNP (-40 mmHg, P5).

of mild LBNP (-20 mmHg). During moderate LBNP (-40 mmHg), there was a significant interaction effect indicating that heart rate increased 6.3 beats/min (95%CI 2.0 to 11; P = 0.005) more than without LBNP. During isometric handgrip, CO was reduced by 18% during mild LBNP (-20 mmHg) and 30% during LBNP (-40 mmHg) compared to isometric contraction alone. The reduction in CO was perfectly compensated for by an increased TPR that facilitated the maintenance and development of the pressor response to isometric exercise.

When preload was transiently reduced during an ongoing isometric handgrip contraction (P2 and P3; **Fig 3** **and Table 2**), we observed a reduction in stroke volume of about 15% during application of mild LBNP (-20mmHg) and 28% during moderate LBNP (-40 mmHg), accompanied by an initial brief drop in MAP. During mild LBNP (-20mmHg) the drop was immediately and solely compensated by a rapid increase in TPR. During moderate LBNP (-40mmHg) the drop was also partly compensated by an increase in heart rate although mainly by a significant increase in TPR during the CO-reduced period.

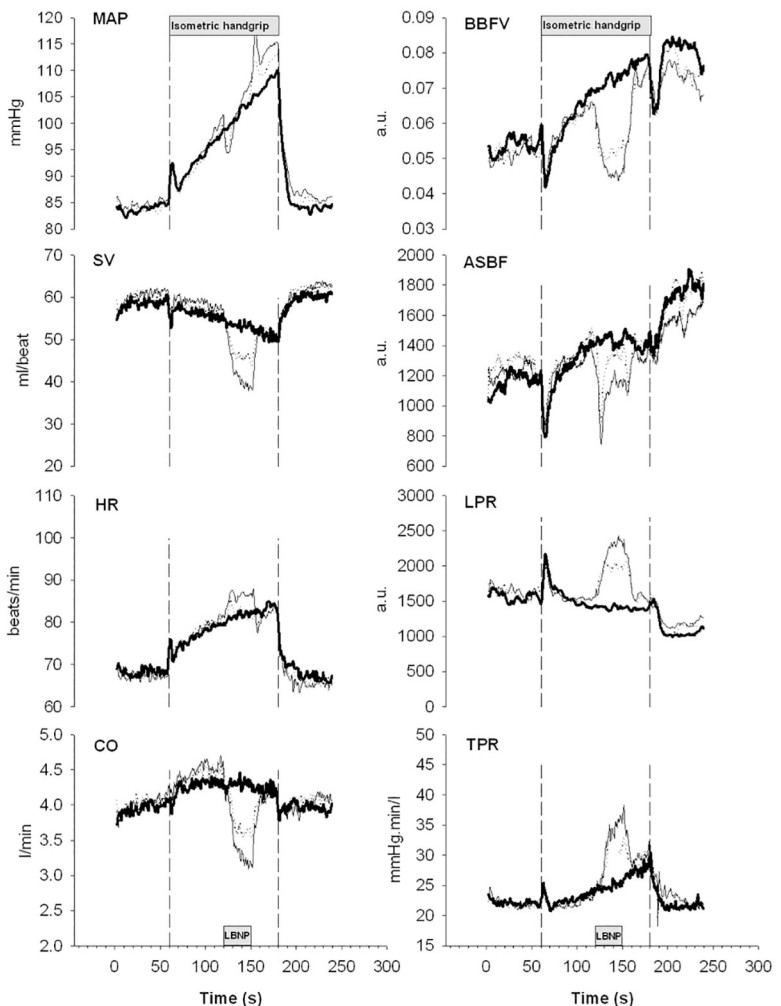

**Fig 3. Pooled average trends of the individual cardiovascular responses (n = 11) to 2 min isometric handgrip contraction (40% of maximal voluntary contraction).** Bold lines show mean arterial pressure (MAP), stroke volume (SV), heart rate (HR), cardiac output (CO), blood flow velocity in the brachial artery (BBFV), acral skin blood flow perfusion (ASBF), local peripheral resistance (LPR) and total peripheral resistance (TPR) during isometric handgrip in supine position (P1). Dotted lines and hairlines show the responses during a transient reduction in preload induced by onset of lower body negative pressure during the ongoing contraction. Dotted lines show the responses to application of mild LBNP (-20mmHg, P2) and hairline the response to moderate LBNP (-40mmHg, P3).

## Discussion

The present study reveals a considerable flexibility in the cardiovascular control mechanisms used to maintain the pressor response to isometric exercise. The main finding is that a gradual increasing TPR is the most important factor contributing to the observed increase in MAP during isometric exercise, both in supine position and during different levels of reduced preload.

### CO during isometric exercise

The results show that in supine position SV gradually decreases at a slope around -4 mL/min during isometric handgrip. This is in accordance to what we previously have recorded [11]. A gradually increasing HR maintains CO, while TPR is the main factor contributing to the

**Table 1. Hemodynamic response to IHG during continuous LBNP.**

| | Estimates (95% CI) | | Main effects | | Interaction effect (IHG: LBNP) |
|---|---|---|---|---|---|
| | **Before IHG** | **During IHG** | **IHG** | **LBNP** | |
| **Mean arterial pressure (mmHg)** | | | | | |
| LBNP 0, P1 | 84 (76 to 93) | 109 (100 to 117) | 24 (18 to 30; P < 0.001) | | |
| LBNP-20, P4 | 85 (76 to 93) | 111 (102 to 120) | | 0.03 (-6.3 to 6.4; P = 0.99) | 2.3 (-6.6 to 11.3; P = 0.61) |
| LBNP-40, P5 | 85 (76 to 94) | 114 (105 to 123) | | 0.4 (-6.0 to 6.8; P = 0.91) | 5.0 (-4.1 to 14.0; P = 0.28) |
| **Cardiac output (l/min)** | | | | | |
| LBNP 0, P1 | 4.0 (3.5 to 4.4) | 4.4 (3.9 to 4.9) | 0.40 (0.16 to 0.64; P = 0.001) | | |
| LBNP-20, P4 | 3.3 (2.8 to 3.8) | 3.6 (3.1 to 4.0) | | -0.69 (-0.93 to -0.46; P < 0.001) | -0.12 (-0.46 to 0.21; P = 0.47) |
| LBNP-40, P5 | 2.9 (2.4 to 3.3) | 3.1 (2.6 to 3.5) | | -1.11 (-1.35 to -0.87; P < 0.001) | -0.21 (-0.55 to 0.13; P = 0.23) |
| **Heart rate (beats/ min)** | | | | | |
| LBNP 0, P1 | 69 (62 to 75) | 84 (78 to 90) | 15.6 (12.6 to 18.6; P < 0.001) | | |
| LBNP-20, P4 | 68 (62 to 74) | 87 (81 to 93) | | -0.6 (-3.6 to 2.4; P = 0.71) | 3.5 (-0.7 to 7.8; P = 0.11) |
| LBNP-40, P5 | 74 (67 to 80) | 96 (89 to 102) | | 5.1 (2.0 to 8.1; P = 0.001) | 6.3 (2.0 to 10.6; P = 0.005) |
| **Stroke volume (ml)** | | | | | |
| LBNP 0, P1 | 59 (52 to 66) | 53 (46 to 60) | -6.7 (-9.9 to -3.5; P <0.001) | | |
| LBNP-20, P4 | 49 (42 to 56) | 42 (35 to 49) | | -10.2 (-13.4 to -7.1; P < 0.001) | -0.6 (-5 to 3.9; P = 0.80) |
| LBNP-40, P5 | 40 (33 to 47) | 34 (27 to 41) | | -19.2 (-22.4 to -16; P = < 0.001) | 0.1(-4.4 to 4.6; P = 0.96) |
| **Total peripheral resistance (mmHg min/ l)** | | | | | |
| LBNP 0, P1 | 21.9 (16.7 to 27.2) | 26.4 (21.2 to 31.7) | 4.5 (1.4 to 7.6; P = 0.005) | | |
| LBNP-20, P4 | 26.4 (21.2 to 31.6) | 32.9 (27.7 to 38.1) | | 4.5 (1.4 to 7.6; P = 0.004) | 2.0 (-2.3 to 6.3; P = 0.37) |
| LBNP-40, P5 | 30.9 (25.6 to 36.1) | 38.9 (33.6 to 44.1) | | 8.9 (5.8 to 12; P < 0.001) | 3.5 (-0.9 to 7.9; P = 0.12) |
| **Local peripheral resistance (a.u.)** | | | | | |
| LBNP 0, P1 | 2298 (1473 to 3124) | 2001 (1175 to 2826) | -298 (-797 to 201; P = 0.24) | | |
| LBNP-20, P4 | 2607 (1779 to 3436) | 2266 (1437 to 3094) | | 309 (-193 to 810; P = 0.23) | -44 (-753 to 665; P = 0.90) |
| LBNP-40, P5 | 2835 (2003 to 3667) | 2636 (1804 to 3468) | | 536 (32 to 1041; P = 0.038) | 99 (-614 to 813; P = 0.79) |

First two columns are estimates and 95% confidence intervals of the first minute (before IHG) and last 30 s of IHG (during IHG) for continuous LBNP 0, -20 and -40 mmHg (P1, 4 and 5), respectively. Last three columns are main effects of IHG (compared to before IHG) and LBNP -20 and -40 mmHg (compared to LBNP 0 mmHg) and their interaction effects with confidence intervals and P-values from the mixed effects regression models.

increase in MAP (**Fig 2**). This is in agreement with some [10–14], while others are referring towards solely CO [5–9] or a contribution of both CO and TPR [15–18]. Interestingly, several studies have depicted that the pressor mechanism during the first minute of handgrip exercise is mainly reliant on CO, whereas that response is blunted afterwards [8, 16]. Similarly, the pressor response switches to TPR if the ability to increase CO is compromised [53–55]. However, the current study shows no increase in CO during the first minute and it also seems

**Table 2. Hemodynamic response to IHG and transient LBNP.**

| | Estimates (95% CI) | | Main effects | |
| --- | --- | --- | --- | --- |
| | **Before IHG** | **During IHG** | **IHG** | **LBNP** |
| **Mean arterial pressure (mmHg)** | 84 (78 to 91) | | 19.5 (15.1 to 23.9; P < 0.001) | |
| LBNP 0, P1 | | 104 (96 to 112) | | |
| LBNP-20, P2 | | 103 (95 to 111) | | -0.7 (-6 to 4.7; P = 0.81) |
| LBNP-40, P3 | | 104 (97 to 112) | | 0.6 (-4.8 to 6.1; P = 0.82) |
| **Cardiac output (l/min)** | 4.0 (3.6 to 4.4) | | 0.4 (0.2 to 0.6; P < 0.001) | |
| LBNP 0, P1 | | 4.5 (4.0 to 4.9) | | |
| LBNP-20, P2 | | 3.7 (3.2 to 4.1) | | -0.8 (-1.0 to -0.5; P < 0.001) |
| LBNP-40, P3 | | 3.3 (2.9 to 3.7) | | -1.2 (-1.4 to -0.9; P < 0.001) |
| **Heart rate (beats/ min)** | 68 (62 to 73) | | 14.8 (12.5 to 17.1; P < 0.001) | |
| LBNP 0, P1 | | 83 (77 to 88) | | |
| LBNP-20, P2 | | 82 (76 to 88) | | -0.5 (-3.3 to 2.4; P = 0.75) |
| LBNP-40, P3 | | 86 (80 to 92) | | 3.5 (0.6 to 6.4; P = 0.018) |
| **Stroke volume (ml)** | 60 (53 to 67) | | -5.6 (-8 to -3.1; P < 0.001) | |
| LBNP 0, P1 | | 54 (47 to 61) | | |
| LBNP-20, P2 | | 46 (39 to 53) | | -8.7 (-11.7 to -5.7; P < 0.001) |
| LBNP-40, P3 | | 39 (32 to 46) | | -15.2 (-18.2 to -12.2; P < 0.001) |
| **Total peripheral resistance (mmHg min/ l)** | 22 (18 to 26) | | 2.5 (-0.2 to 5.3; P = 0.071) | |
| LBNP 0, P1 | | 24 (20 to 29) | | |
| LBNP-20, P2 | | 30 (25 to 35) | | 5.8 (2.4 to 9.1; P < 0.001) |
| LBNP-40, P3 | | 36 (31 to 40) | | 11.3 (7.9 to 14.7; P < 0.001) |

First two columns are estimates and 95% confidence intervals of the first minute (before IHG) and last 20 s of LBNP (during IHG) for intermittent LBNP 0, -20 and -40 mmHg (P1, 2 and 3), respectively. Last two columns are main effects of IHG (compared to before IHG) and LBNP -20 and -40 mmHg (compared to LBNP 0 mmHg) with confidence intervals and P-values from the mixed effects regression models.

unlikely that the CO would be compromised as it addresses healthy and young subjects during moderate intensity exercise. It may be rational to look at differences in the sympathetic nerve activity response to isometric exercise and even though several responder profiles (i.e. positive, negative and non-responders) have been identified, they did not correlate with differences in hemodynamics [56]. The lack of consistency leads one to speculate about individual differences in the mechanism of the pressor response. Watanabe et al. demonstrated a large inter-individual variability in CO and TPR that contributed to the exercise pressor response during isometric exercise [57]. Potential explanations for the large inter-individual variety in the response to isometric exercise include variations in posture during exercise [58], ß-adrenergic receptor activity [59, 60], muscle metaboreflex-mediated cardioaccelerator and peripheral vasoconstriction responses [57], sympathetic outflow direction to vascular beds [61], as well as an offset of peripheral vasoconstriction by ß2-adrenergic vasodilation via circulating epinephrine and local nitric oxide production [62]. Future research is warranted to elaborate on these individual differences in the pressor response mechanism.

## Isometric handgrip and mild LBNP (-20mmHg)

While mild LBNP (-20mmHg) reduces CVP and SV, HR normally is not affected [39, 40]. The results of the present study show that application of mild LBNP (-20mmHg) prior to isometric handgrip reduces SV with about 17% while HR is not affected. MAP is maintained through an increase in TPR. Despite the reduction in SV prior to the isometric handgrip, SV continues to

decrease and HR increases gradually during isometric handgrip, following the same pattern as observed without application of LBNP. The main difference between the series with and without application of mild LBNP (-20mmHg), is a marked reduction in SV and CO, leading to a constantly elevated TPR throughout the period (**Fig 2**). Rapid application of mild LBNP (-20mmHg) during an ongoing contraction leads to a transient drop in SV while HR is not effected, except from a transient increase in the first few seconds, probably due to a startle response [63]. Again, the results show that the drop in MAP due to the reduction in SV is immediately compensated for by a rapid increase in TPR (**Fig 3**).

## Isometric handgrip and LBNP (-40mmHg)

A more marked reduction of CVP during application of LBNP (-40mmHg) normally leads to an increased HR [38–40]. The increased HR maintains CO by compensating the large fall in SV. The results from this study support earlier findings [38–40], as we observed a marked increase in HR during the period when LBNP (-40mmHg) is applied before isometric handgrip. When handgrip was combined with reduced preload induced by LBNP (-40mmHg), HR rose following the same pattern as during LBNP (-40mmHg) alone, but from a higher baseline. But again, the main factor contributing to the gradual increase in MAP is a gradual increase in TPR (**Fig 2**). Despite a transient increase in HR, a rapid onset of LBNP (-40mmHg) during an ongoing contraction leads as expected to a rapid reduction in CO and a transient fall in MAP that within a few seconds is perfectly compensated for by an increased TPR (**Fig 3**).

## The mechanisms behind increased TPR

TPR cannot be recorded directly, but can be calculated provided that perfusion pressure (MAP-CVP) and blood flow (CO) are known. For ethical considerations we did not measure CVP in this study, so TPR was defined as MAP divided by CO (mmHg min/L) where MAP was used as an approximation to the perfusion pressure across the systemic circulation. Levenhagen et al. observed an average CVP of around 4 mmHg in rest, 0 mmHg during mild LBNP (-20mmHg) and -4mmHg during LBNP(-40mmHg) [40]. In the present study, TPR is therefore slightly overestimated in rest, while it is almost correct during mild LBNP (-20mmHg) and underestimated during application of LBNP (-40mmHg). However, in this study, it is the relative changes in TPR between the different situations that are of interest and the underestimation of TPR during LBNP (-40mmHg) will not significantly affect our conclusions.

As an indicator of blood flow to the muscle and skin in the resting arm, we measured the blood flow velocity in the brachial artery (BBFV). Since BBFV in the resting arm gradually increases during isometric exercise in P1, it seems that the part of the vascular bed supported by the brachial artery does not contribute much to the observed increase in TPR. On the other hand, we observed significantly lower BBFV in the resting arm during application of LBNP both prior to (**Fig 2**) and during (**Fig 3**) an ongoing isometric contraction. In the regression model, we found a significant increase in local peripheral resistance in the resting arm during LBNP -40mmHg, but no increase with IHG. This finding indicates that muscles in the resting arm contribute to the increase in TPR and blood pressure in situations where CVP is reduced as a result of the reduced preload. Blood flow in the brachial artery reflects the vascular resistance bought through muscles and skin in the resting arm. To differentiate the potential contribution from the resting arm to the increased TPR during isometric exercise, we measured the blood flow in acral skin (ASBF) in the resting arm in addition to blood flow in the brachial artery. Blood flow in acral skin increased during isometric exercise in the same way as in the brachial artery, indicating that acral skin does not make a major contribution to the increase in TPR during isometric exercise (**Fig 2**). Rapid onset of LBNP was induced during an ongoing

contraction, where we observed a large transient drop in acral skin blood flow, indicating strong transient vasoconstriction. During both mild LBNP (-20mmHg) and moderate LBNP (-40mmHg), the initial vasoconstriction at the onset of LBNP attenuated after a few seconds, and blood flow in acral skin was restored, indicating that this part of the vascular bed does not play a vigorous part in increasing TPR during isometric handgrip (**Fig 3**).

It is believed that the purpose of the nervous response to isometric exercise is to ensure adequate blood flow and oxygen delivery to working muscle, and increasing MAP intuitively seems reasonable to perfuse an isometrically contracting muscle. Increased muscle sympathetic nerve activity (MSNA) is believed to mediate vasoconstriction and reduced conductance in skeletal muscle, and although the mechanisms may differ from that of working muscle, it is also believed to increase in non-contracting muscle [64]. Our results indicate that the muscles and skin in the resting arm do not play an important role in increasing TPR during isometric handgrip in supine position, but do make a contribution during reduced preload caused by application of LBNP. This seems to contradict the notion that increased MSNA leads to increased vascular resistance in non-contracting skeletal muscle. In the present study, we did not measure blood flow in other large arteries, but it may seem that other parts of the vascular bed play the most vigorous role in the upregulation of TPR during isometric handgrip. Further investigations on blood flow and vascular resistance in different vascular beds (e.g. cerebral, splanchnic and renal) during IHG are warranted.

## Methodological considerations

We were not able to perform a sample size calculation, as we did not find adequate data for our research question and study design. The subjects in the present study were a convenience sample with a number within the range of what is usual in this kind of experimental study. Although the number of subjects is low, the within-subject repeats narrowed the confidence intervals of the estimates.

In the present study, we did not measure MSNA which is often performed in mechanistic studies on IHG. However, MSNA is a mediator of the circulatory effects of IHG. As we measured these circulatory effects, we evaluated a more downstream effect of IHG.

## Conclusions

According to our findings, the increased TPR is the most important factor contributing to the increase in MAP during isometric handgrip. The results reveal that MAP is maintained by changes in TPR even in situations where CO is reduced both prior to but also transiently during an ongoing isometric contraction. During isometric handgrip in supine position, the resting arm does not play an important role to increase of TPR. During reduced preload we also observed an increased resistance in the resting arm contributing to the increased TPR. This highlights the relative importance for peripheral resistance, compared to CO in mediating the pressor response during isometric exercise.

## Supporting information

**S1 File.**
(TXT)

**S1 Table.**
(TXT)

**S2 Table.**
(TXT)

## Acknowledgments

The authors would like to thank the participants for their involvement in the study.

## Author Contributions

**Conceptualization:** Jonny Hisdal, Einar Stranden.

**Data curation:** Niels A. Stens, Jonny Hisdal, Espen F. Bakke, Narinder Kaur, Archana Sharma, Einar Stranden, Dick H. J. Thijssen, Lars Øivind Høiseth.

**Formal analysis:** Niels A. Stens, Jonny Hisdal, Espen F. Bakke, Narinder Kaur, Archana Sharma, Einar Stranden, Lars Øivind Høiseth.

**Investigation:** Jonny Hisdal, Einar Stranden.

**Methodology:** Jonny Hisdal, Einar Stranden.

**Project administration:** Jonny Hisdal.

**Resources:** Einar Stranden.

**Software:** Jonny Hisdal, Einar Stranden.

**Supervision:** Jonny Hisdal, Dick H. J. Thijssen.

**Validation:** Jonny Hisdal.

**Visualization:** Niels A. Stens, Jonny Hisdal, Einar Stranden.

**Writing – original draft:** Niels A. Stens, Jonny Hisdal, Espen F. Bakke, Narinder Kaur, Archana Sharma, Einar Stranden, Dick H. J. Thijssen, Lars Øivind Høiseth.

**Writing – review & editing:** Niels A. Stens, Jonny Hisdal, Espen F. Bakke, Narinder Kaur, Archana Sharma, Einar Stranden, Dick H. J. Thijssen, Lars Øivind Høiseth.

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
