## [Decision Letter · Decision Letter 0]

15 Oct 2020

PONE-D-20-26181

The relative importance of central versus peripheral factors in mediating the pressor response to isometric muscle contraction in healthy volunteers

PLOS ONE

Dear Dr. Lars Øivind Høiseth

Thank you for submitting your manuscript to PLOS ONE. After careful consideration, we feel that it has merit but does not fully meet PLOS ONE’s publication criteria as it currently stands. Therefore, we invite you to submit a revised version of the manuscript that addresses the points raised during the review process.

The manuscript has been assessed by one reviewer; their comments are available below.

The reviewer have raised some concerns that need attention in a revision. The reviewer note that Title, introduction and discussion rational sections needs review and they raise questions regarding the conclusion employed. 

We look forward to receiving your revised manuscript.

Kind regards,

Gerson Cipriano Jr., PT, MsC, Ph.D.

Academic Editor

PLOS ONE

Additional Editor Comments:

Dear Dr Lars Øivind Høiseth

I want to ask you also to incorporate some additional information, including:

- Clinical Trial Registration Number;

- Study design description;

- Comprehensive statistical analysis description including normality test,

- within- and between difference and sample size calculation;

- Review of the table's legends, including p-value, units of measure, the measure of central tendency, and dispersion.

- Limitation section including the topics presented by the reviewer.

Journal Requirements:

Reviewers' comments:

Reviewer's Responses to Questions

**Comments to the Author**

1. Is the manuscript technically sound, and do the data support the conclusions?

Reviewer #1: Yes

2. Has the statistical analysis been performed appropriately and rigorously? 

Reviewer #1: Yes

3. Have the authors made all data underlying the findings in their manuscript fully available?

Reviewer #1: Yes

4. Is the manuscript presented in an intelligible fashion and written in standard English?

Reviewer #1: Yes

5. Review Comments to the Author

Reviewer #1: This study examined the potential interaction between the effects of lower-body negative pressure (LBNP), which unloads the low-pressure baroreceptors and causes a sustained increase in muscle sympathetic nerve activity (MSNA), and the effects of isometric handgrip exercise (IHG), which causes a progressive increase in MSNA. The authors have used a good design, applying two different levels of LBNP - either in isolation or during IHG. While the results are clear, I do not agree with the authors use of the term "central" to refer to events in the heart. As such, the title is very misleading.

There is strong evidence that increases in MSNA during IHG are due to increases in sympathetic outflow to muscle vascular bed, leading to an increase in total peripheral resistance (TPR), with peripheral feedback from metaboreceptors coming in later in the contraction as metabolites accumulate. Most investigators consider "central" to mean "central command," which originates in the brain and is responsible for the increase in sympathetic outflow to the blood vessels and the heart, and withdrawal of parasympathetic outflow to the heart, that parallels the increase in skeletomotor and fusimotor activity to the contracting muscles. Indeed, the increase in heart rate is considered to be exclusively due to the increase in central command. Peripheral feedback includes the metaboreceptors and mechanoreceptors in the contracting muscle. Moreover, recent evidence has shown that the increase in MSNA is directed to contracting as well as non-contracting muscle.

The authors' conclusions that an increase in TPR is responsible for the increase in MAP during IHG is not particularly novel: this has been known for a long time. The authors need to state what their observations contribute to the literature, particularly given that they did not record MSNA or central venous pressure.

6. PLOS authors have the option to publish the peer review history of their article (what does this mean?). If published, this will include your full peer review and any attached files.

Reviewer #1: **Yes: **Vaughan G Macefield

---

## [Author Response · Author response to Decision Letter 0]

20 Nov 2020

Lars Øivind Høiseth

Dept. of Anesthesiology

Oslo University Hospital

Oslo

Norway

Gerson Cipriano Jr., PT, MsC, Ph.D. 

Academic Editor PLOS ONE

Oslo 15th November 2020

Response to editor and reviewer, PONE-D-20-26181

In behalf of the authors, I would like to thank the editor and reviewer for the thorough review of our manuscript and for the opportunity to submit a revised version. Below, we have gone through the changes made during the revision.

Academic editor: 

- Clinical Trial Registration Number

Due to the experimental nature of this study, it was not registered in a clinical trial registry. 

- Study design description

The study design has been added to the title, now reading: “Factors mediating the pressor response to isometric muscle contraction: An experimental study in healthy volunteers during lower body negative pressure”

 - Comprehensive statistical analysis description including normality test, - within- and between difference and sample size calculation; 

We have added a sentence regarding normality assumptions (lines 187-189). Within- and between group differences are presented in Tables 1 and 2. When planning the study, we were unfortunately not able to find data allowing for a meaningful sample size analysis. We therefore performed the study with a convenience sample, including a number of subjects within the range which is usual for this kind of experimental study (lines 361-365). 

- Review of the table's legends, including p-value, units of measure, the measure of central tendency, and dispersion. 

We have reviewed the tables´ legends and performed minor revisions. 

- Limitation section including the topics presented by the reviewer.

We have added a section “Methodological considerations”, addressing the above limitations and those presented by the reviewer. 

Reviewer

Reviewer #1: This study examined the potential interaction between the effects of lower-body negative pressure (LBNP), which unloads the low-pressure baroreceptors and causes a sustained increase in muscle sympathetic nerve activity (MSNA), and the effects of isometric handgrip exercise (IHG), which causes a progressive increase in MSNA. The authors have used a good design, applying two different levels of LBNP - either in isolation or during IHG. While the result are clear, I do not agree with the authors use of the term "central" to refer to events in the heart. As such, the title is very misleading. There is strong evidence that increases in MSNA during IHG are due to increases in sympathetic outflow to muscle vascular bed, leading to an increase in total peripheral resistance (TPR), with peripheral feedback from metaboreceptors coming in later in the contraction as metabolites accumulate. Most investigators consider "central" to mean "central command," which originates in the brain and is responsible for the increase in sympathetic outflow to the blood vessels and the heart, and withdrawal of parasympathetic outflow to the heart, that parallels the increase in skeletomotor and fusimotor activity to the contracting muscles. Indeed, the increase in heart rate is considered to be exclusively due to the increase in central command. Peripheral feedback includes the metaboreceptors and mechanoreceptors in the contracting muscle. Moreover, recent evidence has shown that the increase in MSNA is directed to contracting as well as non-contracting muscle. The authors' conclusions that an increase in TPR is responsible for the increase in MAP during IHG is not particularly novel: this has been known for a long time. The authors need to state what their observations contribute to the literature, particularly given that they did not record MSNA or central venous pressure.

We used the terms “central” as opposed to “peripheral” out of our focus on circulation (e.g. total peripheral resistance). We do acknowledge that this is misleading in the context of the present manuscript, and have changed this throughout the manuscript. We now simply refer to the respective variables by their names. 

We also agree with the reviewer regarding the current understanding of the mechanisms behind the nervous responses to IHG. Although it is indeed a limitation to our manuscript that we did not measure MSNA, we believe it is a strength that we measured local blood flow and thereby derived resistance to the non-contracting arm which represents a more downstream mediator of the pressor response. The fact that we did not find increased local resistance in the resting arm during IHG is somewhat surprising, regarding the current understanding as stated by the reviewer, and we have emphasized this finding by adding this variable to Table 1. 

Yours sincerely

Lars Øivind Høiseth, MD, PhD

---

## [Editor Report · Decision Letter 1]

25 Nov 2020

Factors mediating the pressor response to isometric muscle contraction: An experimental study in healthy volunteers during lower body negative pressure

PONE-D-20-26181R1

Dear Dr. Høiseth,

We’re pleased to inform you that your manuscript has been judged scientifically suitable for publication and will be formally accepted for publication once it meets all outstanding technical requirements.

Kind regards,

Gerson Cipriano Jr., PT, MsC, Ph.D.

Academic Editor

PLOS ONE
---

## [Editor Report · Acceptance letter]

1 Dec 2020

PONE-D-20-26181R1 

Factors mediating the pressor response to isometric muscle contraction: An experimental study in healthy volunteers during lower body negative pressure 

Dear Dr. Høiseth:

I'm pleased to inform you that your manuscript has been deemed suitable for publication in PLOS ONE. Congratulations! Your manuscript is now with our production department. 

Kind regards, 

on behalf of

Professor Gerson Cipriano Jr. 

Academic Editor

PLOS ONE